# Distinct cellular and molecular mechanisms contribute to the specificity of the two *Drosophila melanogaster* chitin synthases in chitin deposition

Joan Bertran-Mas[1,☮], Ettore De Giorgio[1,2,☮], Nicolás Martín[1], Marta Llimargas[1,*]

**1** Institut de Biologia Molecular de Barcelona, IBMB-CSIC, Department of Cells and Tissues, Parc Científic de Barcelona, Barcelona, Spain, **2** Institut Curie, PSL Research University, CNRS UMR3215, INSERM U934, UPMC Paris-Sorbonne, Paris, France

☮ These two authors contributed equally to this work.
* mlcbmc@ibmb.csic.es

## Abstract

Chitin is a major component of arthropod extracellular matrices, including the exoskeleton and the midgut peritrophic matrix. It plays a key role in the development, growth and viability of insects. Beyond the biological importance of this aminopolysaccharide, chitin also receives considerable attention for its practical applications in medicine and biotechnology, as it is a superior biopolymer with excellent physicochemical and mechanical properties. Chitin is synthesised and deposited extracellularly by chitin synthases. Most insects encode two types of chitin synthases: type A, which are presumed to be required for exoskeleton formation, and type B, which are thought to produce the peritrophic matrix. However, the factors that contribute to the specificity of each type of chitin synthase remain unclear. Here, we leverage the advantages of *Drosophila melanogaster* for functional manipulations to evaluate the mechanisms of activity and the functional requirements of Kkv (Chitin synthase A) and Chs2 (Chitin synthase B). We first demonstrate that Chs2 is expressed and required in a specific region of the larval proventriculus responsible for producing chitin in the peritrophic matrix. We then assess whether the two chitin synthases can functionally substitute for each other. Additionally, we examine their subcellular localisation in different tissues and their ability to deposit chitin in combination with known auxiliary proteins. Our results indicate that these two different chitin synthases are not functionally interchangeable and that they use distinct cellular and molecular mechanisms to deposit chitin. We suggest that the specificity of insect chitin synthases may underlie the production of chitin polymers with different properties, conferring different physiological activities to the extracellular matrices.

**Data availability statement:** All relevant data are within the manuscript and its Supporting Information files.

**Funding:** This study was supported by the Ministerio de Ciencia e Innovación (MCINN) PGC2018-098449-B-I00 and PID2021-126689NB-I00 to ML; Pre2022-102439 to JBM and BES-2016-076723 to EDG. The funders had no role in study design, data collection and analysis, decision to publish, or preparation of the manuscript.

**Competing interests:** The authors have declared that no competing interests exist.

## Author summary

In this study, we investigated the specificity of the two chitin synthases in *D melanogaster*, named Kkv and Chs2. Chitin synthases are enzymes that produce chitin, an abundant polysaccharide found in many organisms, including arthropods and fungi. In insects, chitin, which is essential for development and viability, is the principal component of the exoskeleton and is also part of a layer lining the gut. In insects, chitin is synthesised by two types of enzymes: Chitin synthase A (Kkv), thought to be responsible for chitin deposition in the exoskeleton, and Chitin synthase B (Chs2), believed to mediate chitin deposition in the gut. Whether each type of chitin synthase uses specific mechanisms to deposit chitin has not been addressed. Here we demonstrate that Chs2 localises to the most anterior region of the midgut and is responsible for gut-specific chitin deposition. Importantly, we show that the two enzymes cannot replace each other and that they rely on distinct cellular and molecular mechanisms to perform their role. Beyond its biological significance, chitin has broad applications in biotechnology and medicine as a high-performance biomaterial. Thus, our analysis may inspire new uses of chitin in biotechnology or inform novel strategies to target insect development.

## Introduction

Chitin is an abundant polymer in arthropods and represents a principal component of the apical extracellular matrix, which provides structural and mechanical strength and protects against dehydration, injuries and pathogens. It is found in the exoskeletons, tracheal and internal tendon cuticles, and in the peritrophic matrix (PM). Chitin is composed of linear polymers of β-(1–4)-linked N-acetylglucosamines (GlcNAc) that organise in microfibrils. The chitin chains in the microfibrils can adopt parallel or antiparallel orientations, giving rise to different allomorphs known as α-chitin (antiparallel), β-chitin (parallel) and γ-chitin (parallel and antiparallel). α-chitin forms a highly ordered crystalline structure, while the β- and γ-chitin allomorphs are less tightly packed, more flexible and hydrated [1–5]. Besides its physiological activities, chitin and its deacetylated form, chitosan, are considered most promising biopolymers and serve multiple applications in biotechnology, biomedicine, agriculture, food industry or cosmetics [6–8]. Thus, there is a growing interest in understanding the mechanisms of chitin synthesis and deposition to better leverage the excellent properties of this biomaterial [9].

Chitin is synthesised by a dedicated type of enzymes known as chitin synthases (CHS) that belong to the β-glycosyltransferase family, which polymerise and extrude the polymer to the extracellular space [10]. Insects typically encode two different types of CHS, which belong to the class-A and class-B. It is assumed that class A are expressed and produce chitin in ectodermal tissues while class B are expressed and synthesise chitin in the gut PM [1,3–5,11]. The PM consists of a layer of chitin and glycoproteins that lines the midgut and is required for digestion and protection against pathogens, abrasion and toxins. Depending on the intestinal region that secretes it, the PM is classified as Type I or Type II. Type I PM is typically produced

along the length of the midgut, which generate delaminating chitin lamellae. In contrast, Type II PM is typically produced by a specialised tissue at the anterior region of the midgut, the cardia, from where it is secreted into the midgut. Many previous studies have classified different insect orders as producers of Type I (e.g. Coleoptera, Odonata, Lepidoptera) or Type II (e.g. primitive orders and dipteran larvae) PMs, however, various variations and hybrid mechanisms have been described in different species [12–14].

Knock-down studies in different insect species have identified specific functional requirements for class A and class B CHS, requirements that may respond to their specific pattern of expression in ectoderm or gut, respectively (reviewed in [3,4]). In agreement with this, in *Tribolium castaneum*, CHS1 (class A) RNAinterference affected molting and decreased chitin content in the whole-body, whereas CHS2 (class B) RNAinterference affected feeding and larval size and specifically decreased chitin content in the midgut [15]. However, the previous studies have not addressed whether the two different types of insect CHS are functionally equivalent or interchangeable, and whether besides their expression pattern, other factors contribute to their specificity. Here we take advantage of *D. melanogaster* as an ideal model for functional analysis and ask about the specificity and mechanisms of activity of insect class A and class B CHS.

*D. melanogaster* encodes *krotzkopf verkehrt (kkv)* and *Chitin Synthase 2* (*Chs2*), which belong to the class A and class B CHS, respectively [16,17]. Kkv has been largely studied and there is currently extensive information about the functional requirements, the pattern of expression, subcellular localisation and its molecular mechanism of activity [18–22]. Kkv is expressed and required for chitin deposition in the ectoderm (epidermis and trachea). It localises at the apical membrane, where it polymerises chitin polymers that are extruded to the extracellular space. The ability of Kkv to translocate the polymers and deposit them extracellularly depends on the activity exerted by two interchangeable proteins, Expansion (Exp) and Rebuf (Reb) [20]. In the absence of Exp/Reb, Kkv can polymerise chitin but it cannot translocate it extracellularly. We recently hypothesised that Exp/Reb regulate the conformational organisation of Kkv at the membrane regulating in this way the translocation activity [19]. Chitin deposition by Kkv is assisted by additional auxiliary proteins. Some of these, including SERCA, Ctl2 and Fabp [23–25], have been shown to physically interact with Kkv and are proposed to regulate Kkv trafficking from the ER to the plasma membrane. The nascent chitin microfibrils synthesised by Kkv interact with a protein complex, which includes Obst-A, Rtv and Knk [26–28], that protects and organises the chitin matrix. This chitin matrix is further remodelled by the activity of chitin deacetylases and chitinases [29,30].

In sharp contrast to Kkv, very little information is available on Chs2. Kkv and Chs2 share common domains, like several transmembrane domains, the conserved catalytic domain QRRRW, and a WGTRE motif, but differ in the presence of a coiled coil domain at the C terminal region in Kkv, which is absent in Chs2 [31]. According to Flybase, Chs2 is mainly expressed in intestinal tissues and is predicted to synthesise chitin, presumably deposited in the PM. A recent study documented the expression of *Chs2* in a specific region of the proventriculus in the adult [32], which was proposed to synthesise the Type II PM in the fly [33,34]. However, no information is available on the functional requirements, activity, or mechanisms of chitin deposition employed by Chs2. We also do not have any information about possible auxiliary proteins that assist Chs2 in chitin deposition.

Here we asked whether Kkv and Chs2 are functionally equivalent and if they use a comparable mechanism of activity to synthesise chitin. To this end, we have first characterised chitin deposition in the PM and the functional requirements of Chs2 in this process. We have then attempted to replace each CHS with the other one. Finally, we have investigated whether the cellular and molecular mechanisms of chitin deposition by Kkv are shared with Chs2. Our results indicate distinct mechanisms of regulation and activity for each CHS.

## Results

### 1. Chitin accumulation in the intestinal tract

The digestive tract is divided into three main regions: the foregut (which includes the pharynx, esophagus and part of the proventriculus), the midgut (central part of the gut) and the hindgut (starting at the midgut-Malpighian tubules junction)

[35], https://flygut.epfl.ch/; (Fig 1A). The digestive tract is lined by a PM organised into different layers, L1-L4, which are visible using electron microscopy [33]. *D. melanogaster* larvae are proposed to produce Type II PM, which is secreted in the proventriculus and slides posteriorly to the midgut enclosing the passing food bolus [12,34]. Chitin is one of the components of the PM, presumably enriched in its L2 layer, which helps to organise the glycoprotein mesh that lines the gut [12,13,32,33]. We first sought to find a reliable fluorescent probe to visualise chitin "in situ" in the PM.

A fluorescent-CBP (chitin binding protein) probe, which consists of the chitin binding domain of the Chitinase A1 from *Bacillus circulans,* has been routinely used to visualise chitin in ectodermal tissues (e.g. trachea, epidermis) [36]. CBP staining, also used "in vivo" [37], nicely and cleanly reveals endogenous and ectopic (intracellular or extracellular) chitin synthesised by Kkv [19]. We stained the gut of late L3 larvae and examined the presence of CBP in the intestinal tract.

As the PM-chitin is synthesised in the proventriculus [34], we paid special attention to this region. The proventriculus is formed by the invagination of the esophagus into the midgut, which generates a bulb-shaped structure with 3 layers [33,34] (Fig 1B). As described by [33,34], the larval proventriculus has been divided into different regions according to the cell morphology and cell type (Figs 1C and S1A). A group of cells in the outer layer (PR cells) were proposed to produce the PM chitin. PR cells, as well as cells anterior (APR) and posterior to PR (PPR), correspond to the most anterior midgut cells, while cells anterior to APR correspond to ectodermal foregut cells [34]. We detected a dense, fibrous and strong accumulation of CBP lining the lumen of the esophagus and the lumen of the proventriculus (Fig 1D, 1E). In addition, the region in the proventriculus that corresponds to PR cells displayed a distinct enrichment of chitin (Fig 1D, 1E, 1G, yellow arrow, and S1B). We also detected the presence of chitin intracellular punctae in the PR cells (Figs 1D–1F, blue arrow). By using specific ectodermal and endodermal markers, particularly Forkhead (Fkh, ectoderm) and Defective proventriculus (Dve, endoderm) [38–40]) we confirmed CBP staining along the apical domain of the ectodermal region and its enrichment along the apical domain of endodermal cells (S1C and S1D Fig).

Along the midgut region, we detected a continuous, thin and faint layer that lined the gut epithelium (Fig 1I', yellow arrow and inset). In addition, we also detected a homogeneous pattern of CBP staining in the midgut lumen that suggests accumulation of chitin material along the digestive tract (Fig 1H' orange arrow).

In the most posterior region, the hindgut, we detected a thick and compact CBP staining (Fig 1J).

## 2. Chs2 deposits chitin in the PM

We hypothesised that the chitin detected in the esophagus and anterior region of the proventriculus corresponds to the cuticle, likely synthesised by Kkv in foregut ectodermal cells. In contrast, the chitin enrichment observed in the lumen of the proventriculus likely corresponds to chitin deposited in the nascent PM synthesised by endodermal PR cells expressing Chs2 [34], Fig 1B, 1C, and S1A-S1D). Consistently, recent data from single-cell RNA sequence analysis indicated that Chs2 is specifically expressed in the corresponding region of the adult proventriculus [32].

To test this, we aimed to analyse the contribution of *Chs2* to chitin deposition in the digestive tract, as no functional analysis of *Chs2* has been performed to date in *D. melanogaster*. We found that a combination of deficiencies removing Chs2 was not viable; however, we were able to obtain L3 larval escapers (S1F Fig). Analysis of ectodermal and endodermal markers indicated that the proventriculus was correctly organised in these escapers (S1E Fig). However, when analysing chitin deposition, we observed a clear difference compared to the control: escapers showed accumulation of chitin in the esophagus and anterior region of the proventriculus, but lacked enrichment in the PR region (Figs 1K–1N, S1E, and S1F). Additionally, the layer lining the midgut cells was absent in L3 escapers (Figs 1P, 1P', and S1F). This phenotype was fully penetrant in L3 larval escapers (n = 12/12 *Df(Chs2)* guts showed no PR chitin enrichment and no layer in the midgut, whereas n = 9/9 control guts showed PR chitin enrichment and chitin layer in the midgut). In contrast, we still detected the homogenous luminal accumulation, indicating that this pattern within the digestive material does not correspond to the chitin synthesised by Chs2 in the PM (Fig 1O). Finally, no differences were observed in the pattern of chitin accumulation

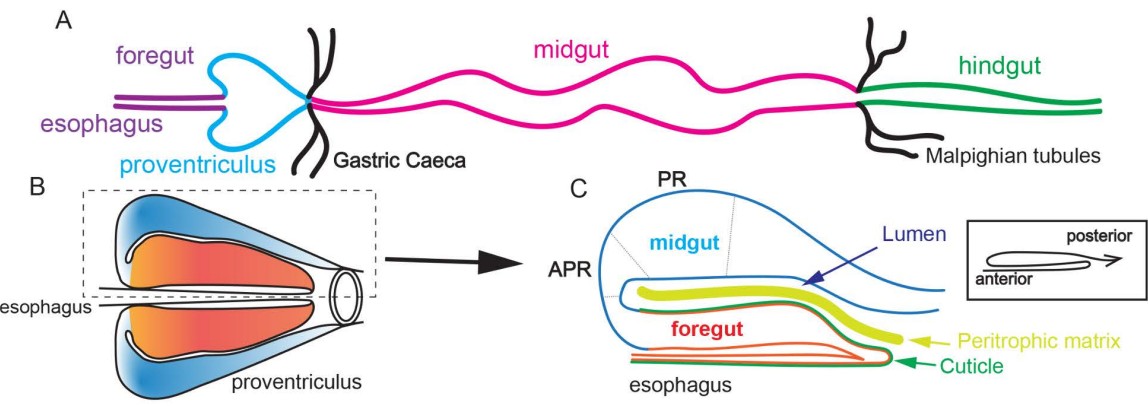

**Fig 1. CBP staining along the larval digestive tract.** (A-C) Schematics of the larval digestive tract (A), proventriculus (B) and sagittal section of half proventriculus (C, corresponding to the boxed region in B). The proventriculus forms at the anterior region of the digestive tract and is composed of the esophagus in the inner wall, foregut (ectodermal) cells (orange region in B,C) in the middle wall, and midgut (endodermal) cells (blue region in B,C)

in the outer wall. The esophagus and foregut cells are lined by cuticle (green in C). The PM (yellow in C) is deposited in the proventricular lumen and moves posteriorly towards the midgut region (magenta in A). Different cell types, i.e PR, APR among others, have been identified in the proventriculus region [34]. (D-Q) L3 digestive tracts stained for chitin using CBP (magenta) and Dlg or α-Spec as cellular markers (green). Images show single sections (E-G, J, L-N,Q), projections of 2–3 sections (H,I,O,P) or a projection of several sections (D,K). In the wild type, chitin is deposited in the esophagus (purple arrows), anterior part of the proventriculus (white arrows), enriched in the proventricular lumen (yellow arrow) and in the PR cells (blue arrows) (D-G). In L3 Df(Chs2) escapers the enrichment in the PR region or cells is not detected (K-N). In the midgut region, a continuous layer along the midgut cells is observed in control (H-I') but not in L3 escapers (O-P') (yellow arrows and magnification of boxed regions in grey). Note the presence of unspecific CBP staining in the lumen of the gut in control and escapers (orange arrows). No differences are detected in the hindgut (ectodermal) region between control and L3 escapers in chitin deposition (J,Q). tr, trachea. Scale bars D,E,H,J,K,L,O,Q 50 µm; F,G,I,M,N,P 20 µm.

in the hindgut (Fig 1Q). Altogether, our findings suggested that the chitin enrichment in the PR region and the chitin layer along the midgut correspond to chitin deposited in the PM by Chs2.

To confirm that the absence of *Chs2* was responsible for the abnormal chitin pattern observed in the *Df(Chs2)* combination, we sought to rescue the phenotype through specific expression of *Chs2*. To this end, we generated *Chs2GFP* and *Chs2* transgenes and expressed them specifically in the PR cells using a Gal4 line [32], S2A Fig) in a *Df(Chs2)* mutant background. Chs2 expression was able to restore detectable chitin accumulation in the PM both in the proventriculus and along the midgut region (Figs 2A–2D and S2C-S2F) (all *PRGal4-UASChs2;Df(Chs2)* guts displayed chitin in the PR region and midgut, n = 8/8, whereas none of the siblings carrying only *Df(Chs2)*, n = 0/8, did). Given that the *PRGal4* line we used shows expression exclusively in the PR cells (S2A Fig) and [32], this result indicates that chitin deposition in this region is sufficient to generate the PM-associated chitin lining the entire intestinal tract. To assess whether Chs2 expression in endodermal cells other than PR could restore chitin deposition, we expressed Chs2 in the midgut (but not in the proventriculus, S2B Fig) using the *Mex1Gal4* driver. We found no chitin deposition in the proventriculus or in the layer along the midgut (S2G and S2H Fig) (n = 9/9 *Mex1Gal4-UASChs2;Df(Chs2)* guts showed no chitin in the PR region and midgut). Altogether these results strongly suggest that, at least at larval stages, PM-associated chitin is synthesised exclusively in PR cells.

Overall, we concluded that Chs2 synthesises chitin deposited in the narrow proventricular lumen of the PR region and in the chitinous layer along the midgut. In addition, our approach identified a reliable and convenient marker to visualise chitin in the PM, CBP.

### 3. Kkv cannot replace Chs2 and deposit chitin in the PM

Kkv deposits chitin in ectodermal tissues [21]. We asked whether it could also deposit chitin in the PM. To test this, we expressed *kkv* in PR cells in a *Df(Chs2)* mutant background. Chitin was not detected in the PR region or in the layer along the midgut. Instead, we detected chitin deposited as intracellular punctae in PR cells (n = 5/5 *PRGal4-UASKkv;Df(Chs2)* guts showed no PR chitin enrichment, no layer in the midgut and intracellular chitin) (Fig 2E, 2F). This pattern indicated the ability of Kkv to polymerise chitin in the PR region but not to translocate it extracellularly.

We previously showed that chitin translocation absolutely depends on the presence of Exp/Reb [19]. Therefore, we co-expressed *reb* and *kkv* in PR cells in a *Df(Chs2)* mutant background. Again, we observed no chitin deposition in the PM of the proventriculus and midgut, and abundant intracellular chitin punctae (n = 11/11 *PRGal4-UASKkv + reb;Df(Chs2)* guts) (Fig 2G, 2H).

Altogether these results show that Kkv cannot replace Chs2 PM-associated chitin deposition, even in the presence of Reb.

### 4. Chs2 cannot replace Kkv and deposit chitin in ectodermal tissues

During embryogenesis, Kkv deposits chitin in the epidermal cuticle and in the trachea [21,41]. In the trachea, chitin is first deposited inside the lumen forming a filament, and at later stages it is also deposited in the procuticle layer of the tracheal

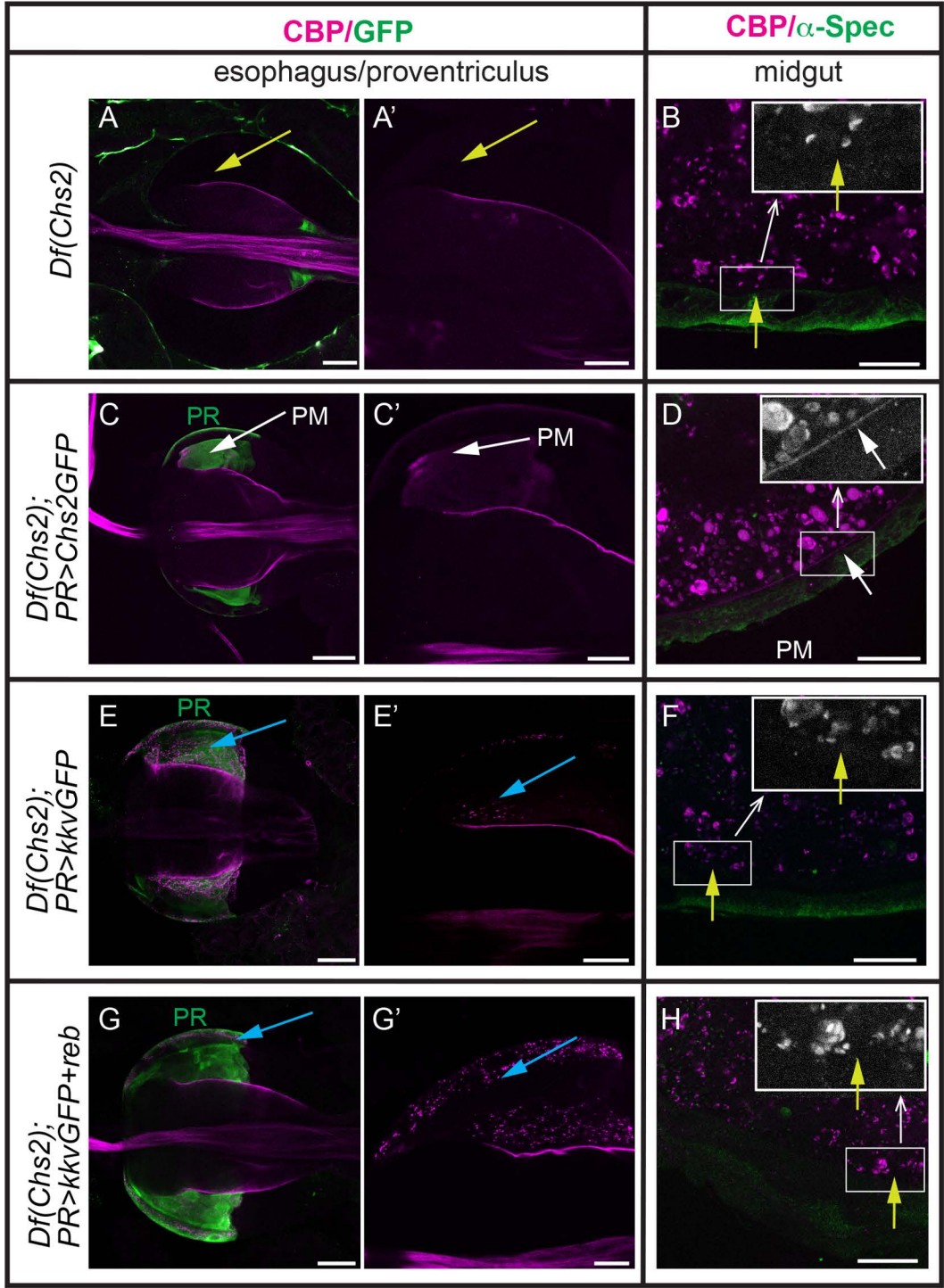

**Fig 2. Chs2, but not Kkv, deposits chitin in the PM.** (A-C) L3 digestive tracts stained for chitin using CBP (magenta) and α-Spec (A,B,D,F,H) or GFP (C,E,G) in the indicated genotypes. All images correspond to single confocal sections. Note the absence of chitin in the PM in L3 *Df(Chs2)* escapers (yellow arrows in A,B) and the rescue when *Chs2GFP* is expressed in PR cells (white arrows in C,D). Expression of *KkvGFP* alone (E-F) or in combination with *reb* (G-H) in PR cells results in chitin accumulation in intracellular punctae (blue arrows in E,G) but no chitin deposition in PM (yellow arrows in F,H). To aid visualisation of the PM layer along the midgut the boxed regions are shown magnified in grey. Scale bars A,C,E,G 50 μm; A',B,C',D,E',F,G',H 20 μm.

cuticle [42,43] (Fig 3A, 3B). In *kkv* mutants, the tracheal chitin filament does not form and the procuticle layers are devoid of chitin [20–22,42,43] (Fig 3C, 3D). The expression of *kkv* in tracheal cells of *kkv* mutants restores the presence of chitin in the luminal filament and in the tracheal cuticle (n = 27/27 *kkv* mutant embryos expressing *kkvGFP* in tracheal cells deposit chitin) (Fig 3E, 3F) [20]. We asked whether Chs2 can substitute for Kkv activity and thus we expressed *Chs2* in the trachea of *kkv* mutants. We found no rescue of chitin accumulation in the filament, in the cuticle or inside the cells (n = 0/16 *kkv* mutant embryos expressing *Chs2GFP* in tracheal cells deposit chitin) (Fig 3G, 3H, and S3A-S3D).

We have previously shown that in combination with the activity of Exp/Reb, Kkv is able to polymerise and deposit chitin extracellularly in ectodermal tissues, even in those that normally do not synthesise this polysaccharide [20]. In the absence of Exp/Reb, Kkv can polymerise chitin but it cannot translocate and deposit it extracellularly, and as a consequence chitin accumulates intracellularly as membrane-less punctae [19]. Thus, we observed intracellular chitin punctae when only *kkv* is expressed in the salivary glands and large amounts of chitin deposited in the lumen when *kkv* and *reb* are co-expressed there [20] (Fig 3I, 3J). We tested the ability of Chs2 to polymerise and deposit chitin when it is expressed in the salivary glands, alone or in combination with Reb. In the absence of Reb, we did not detect accumulation of chitin punctae intracellularly (Figs 3K and S3E). When we co-expressed *Chs2* and *reb* we could not detect increased chitin deposition in the lumen of salivary glands, or intracellularly (Figs 3L and S3F) compared to *reb* expression alone (Fig 3N). While salivary glands do not normally deposit chitin (Fig 3M), we note that the expression of *reb* alone in salivary glands promotes some deposition of chitin in the lumen (Fig 3N), due to the presence of Kkv (S3G and S3H Fig).

To further investigate possible functional requirements of Exp/Reb in Chs2 activity we co-expressed RNAi for *exp* and *reb* in PR cells, where Chs2 is normally active. We found clear deposition of chitin in the PM (n = 11/11 *PRGal4-UASexp RNAi;UASrebRNAi* guts showed PR chitin enrichment, chitin layer in the midgut and intracellular chitin) (S3I and S3J Fig). Altogether our results indicate that Chs2 cannot replace Kkv function, even in the presence of Exp/Reb activity, and that Exp/Reb function is not required for normal Chs2 activity.

### 5. Expression of *Chs2* and *kkv* in the proventriculus

To better understand the functional specificity of each CHS we investigated their endogenous expression patterns. *kkv* is known to be expressed in ectodermal tissues in the embryo [19]. We examined its pattern in the proventriculus using a fluorescently tagged knock-in allele [18]. We found accumulation of Kkv in the apical domain of the most anterior region of the proventriculus, which likely corresponds to the ectodermal region (Fig 4C, 4D). We confirmed the ectodermal nature of Kkv pattern in the proventriculus by the lack of co-staining with the endodermal marker Dve (Fig 4A, 4B, 4E, 4F)

To investigate the in situ expression pattern of *Chs2* in the larval proventriculus, we tested the usefulness of a *Chs2* allele from the CRIMIC collection, *Chs2^CR60212-TG4.0* (Flybase and [44]). This allele contains a *Trojan-GAL4* gene trap in the third intron, which, if inserted in the correct orientation, should result in the expression of GAL4 under the control of the Chs2 regulatory sequences. We found that when crossed to a reporter line (e.g., *UAS-GFP* or *UAS-nlsCherry*), this line reproduces a pattern in the PR region of the proventriculus (Fig 4G, 4J), reflecting the requirement of Chs2 (Fig 2C, 2D, S2E, and S2F). Co-staining with endodermal and ectodermal markers, Dve and Fkh respectively, confirmed *Chs2* expression in the endodermal region of the proventriculus (Fig 4I, 4J), and no expression in the ectodermal region (S4A Fig). Our results closely align with a recent scRNAseq analysis that indicated *Chs2* expression in cells in the adult proventriculus corresponding to PR cells [32]. We also analysed *Chs2* pattern at embryonic stages, and we detected expression in a group of cells at the embryonic proventriculus and an unspecific pattern inside the midgut lumen (S4B and S4C Fig), but no expression in epidermis or trachea.

Altogether this analysis confirmed the specific expression of Kkv in the ectodermal region and Chs2 in the anterior endodermal region in the proventriculus, consistent with the functional requirements for chitin deposition in the cuticle and PM, respectively.

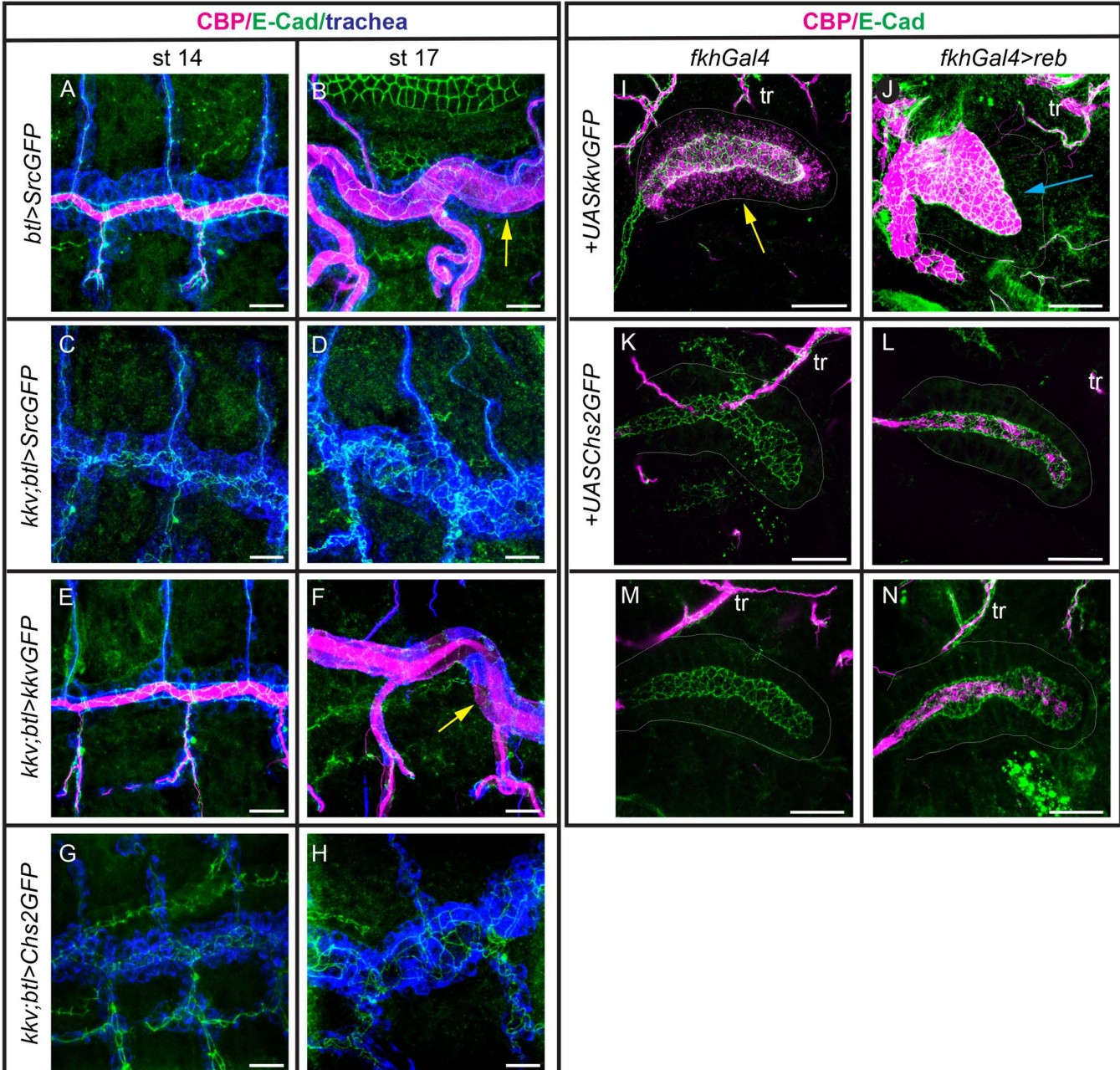

**Fig 3. Kkv, but not Chs2, deposits chitin in the trachea and salivary glands.** (A-H) Confocal projections showing dorso-lateral views of the trachea stained for chitin (CBP, magenta), GFP (blue) and E-Cad (green) to follow the tubes. Note the presence of a luminal filament at early stages (A) and the cuticle with the typical taenidial pattern (yellow arrow in B) in the wild type. No chitin in filament or cuticle is deposited in *kkv* mutants (C,D). Chitin deposition is rescued when adding back *kkv* (E,F), but not Chs2 (G,H). (I-N) Confocal projections of salivary glands stained for chitin (CBP, magenta) and E-Cad (green). Note the capacity of *kkv* to polymerise chitin and deposit it intracellularly (I) or in the lumen of the gland when coexpressed with *reb* (J). In contrast, *Chs2* cannot deposit chitin intracellularly when expressed alone (K). In combination with *reb* there is some deposition of chitin in the lumen (L). This is also observed when *reb* is expressed alone (N), in contrast to control salivary glands (M), and is due to low levels of expression of *kkv* in the salivary glands (see S3H Fig). tr, trachea. Scale bars 10 µm.

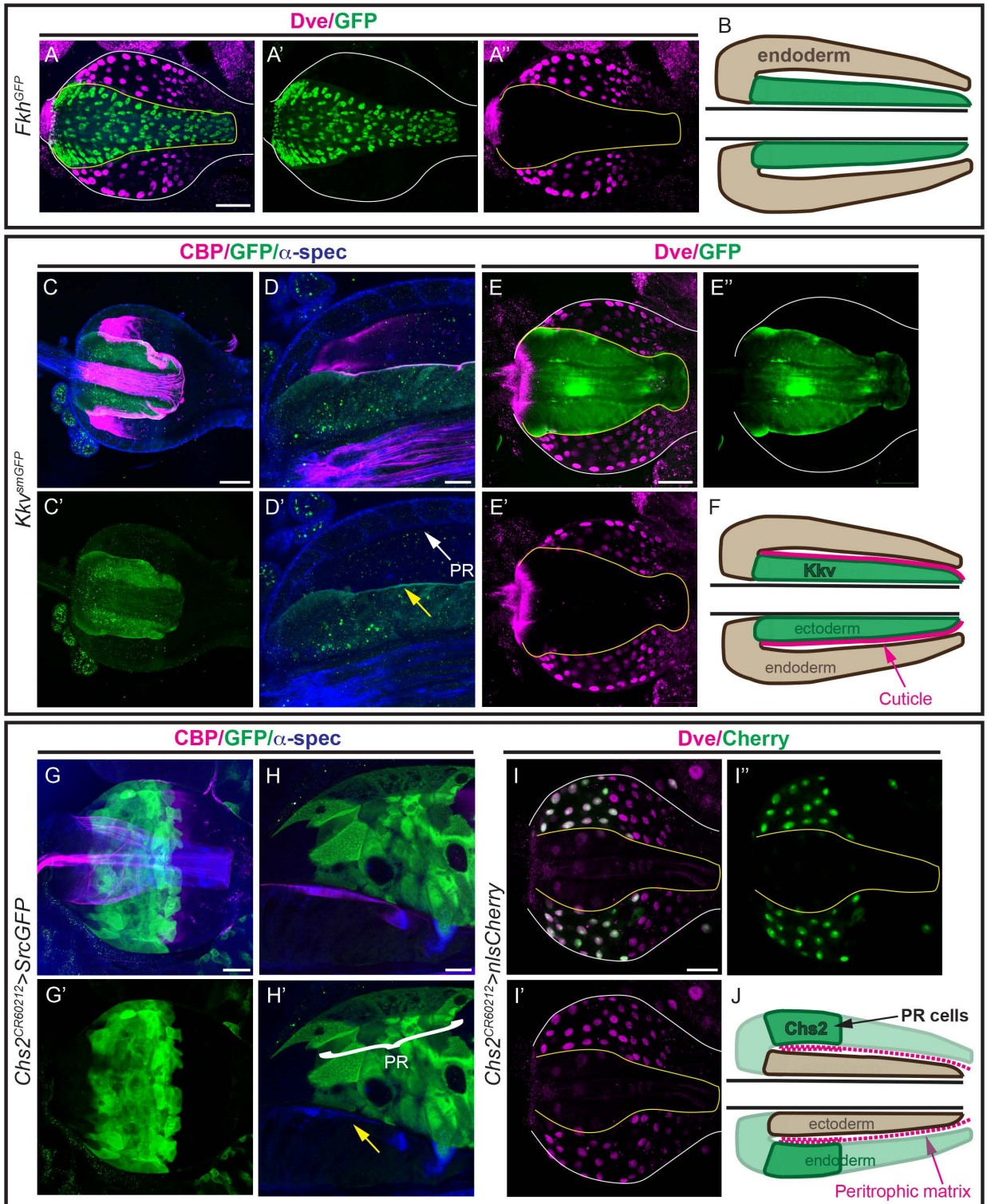

**Fig 4. Expression pattern of Kkv and Chs2 in the proventriculus.** Single confocal sections or projections of 2–3 sections of the proventriculus stained for the indicated markers in the indicated genotypes. (A) Fkh and Dve staining delineate the ectodermal and endodermal regions, respectively, within the proventriculus. A schematic representation is shown in B. (C-F) *kkv* expression is visualised in proventriculus carrying a GFP-tagged knock-in

kkv allele using the GFP pattern (green). Kkv is detected in the ectodermal region adjacent to the cuticle (C), enriched apically (yellow arrow in D'), absent in PR cells (D) and does not colocalise with dve (E). A schematic representation of the Kkv expression pattern is shown in F. (G-J) *Chs2* pattern is visualised in green leveraging Gal4 expression under the control of Chs2 regulatory sequences in the *Chs2CR60212-TG4.0* allele crossed to *UASsrcGFP* or *UASnlsCherry*. *Chs2* is detected in the PR cells (G,H), where it colocalises with dve (I), and is absent from the ectodermal region (yellow arrow in H'). A schematic representation of *Chs2* pattern is shown in J. Scale bars A,C,E,G,I 50 μm; D,H 20 μm.

## 6. Subcellular localisation of Chs2 in endogenous and ectopic tissues

Analysis of *kkv* and *Chs2* expression revealed their specific patterns in ectoderm and endoderm, respectively. Functional analysis revealed that *kkv* cannot replace *Chs2* and *Chs2* cannot replace *kkv*. To better understand this inability of each CHS to compensate for the other, we examined their subcellular localisation.

We first investigated the localisation of Chs2 by expressing *Chs2GFP* in PR cells. We detected a generalised pattern in the cell, along with a conspicuous apical accumulation of the protein, which correlates with chitin deposition in the proventricular lumen (Figs 5A–5C and S5A). No apparent differences in Chs2GFP localisation were observed in the presence of Reb (S5B Fig). We then expressed *Chs2GFP* in other regions of the proventriculus where *Chs2* is not endogenously expressed. Strikingly, while we observed the general pattern in the cell, we could not detect a distinct apical localisation outside the PR region (Fig 5D).

To further analyse Chs2 localisation in ectopic tissues we expressed *Chs2GFP* in various embryonic tissues. In the ectoderm, like trachea, epidermis and salivary glands, we observed that Chs2GFP largely accumulated throughout the cell, without detectable apical enrichment. The cytoplasmic pattern of Chs2GFP reminded the organisation of the endoplasmic reticulum (ER), and double staining with the ER marker KDEL indicated strong colocalisation (Fig 5E, 5F, S6A, and S6B). We did not detect presence of Chs2GFP vesicles in the cytoplasm (Fig 5E, 5F, S6A, and S6B), which likely indicates a lack of intracellular trafficking. While we cannot completely rule out that a portion of the protein exits the ER, our results suggest that Chs2GFP is predominantly retained in the ER.

We aimed to confirm that this ER pattern reflected the localisation of Chs2 protein and was not caused by the presence of the GFP in Chs2GFP. To address this, we generated an antibody against Chs2. Unfortunately, the antibody did not recognise Chs2 endogenous expression in the proventriculus (likely due to low endogenous Chs2 levels combined with the antibody's limited efficiency), however we found that it recognised Chs2 when overexpressed. Thus, we missexpressed full-length untagged Chs2 in ectodermal tissues and analysed its subcellular localisation (S5C and S5D Fig). Co-localisation analysis with membrane (srcGFP) and ER (KDEL) markers revealed that Chs2 is primarily accumulated in the ER when expressed in ectodermal tissues (average Pearson's correlation coefficient, r, for srcGFP/Chs2 was 0,064, while the average for KDEL/Chs2 was 0,693; n = 36 cells from 9 different embryos) (Fig 5H). Our analysis supports the hypothesis that Chs2 is primarily retained in the ER when expressed in non-endogenous tissues, likely unable to reach the membrane.

We aimed to determine whether Chs2 was specifically retained in the ER or whether the accumulation was due to a defective folding. Missfolded proteins are typically degraded from ER upon ubiquitination [45]. FK2 recognises mono and polyubiquitinated conjugates [46]. We found no colocalisation between FK2 and Chs2 (S5F Fig), strongly suggesting that Chs2 protein is retained in the ER by a specific mechanism/s.

As we have found that Chs2 is endogenously expressed and required in PR endodermal cells where it localises apically, we speculated that it could also localise apically in embryonic endodermal cells. However, we observed that Chs2 also localised in the cytoplasm in the embryonic midgut, and KDEL co-staining indicated accumulation in the ER. We did not detect apical enrichment or presence of Chs2 vesicles in the cytoplasm (Figs 5G, S6E, and S6G). The subcellular localisation observed correlated with the inability of Chs2, alone or in combination with Reb, to deposit chitin in the embryonic gut (a tissue that does not deposit detectable chitin) (S6G Fig).

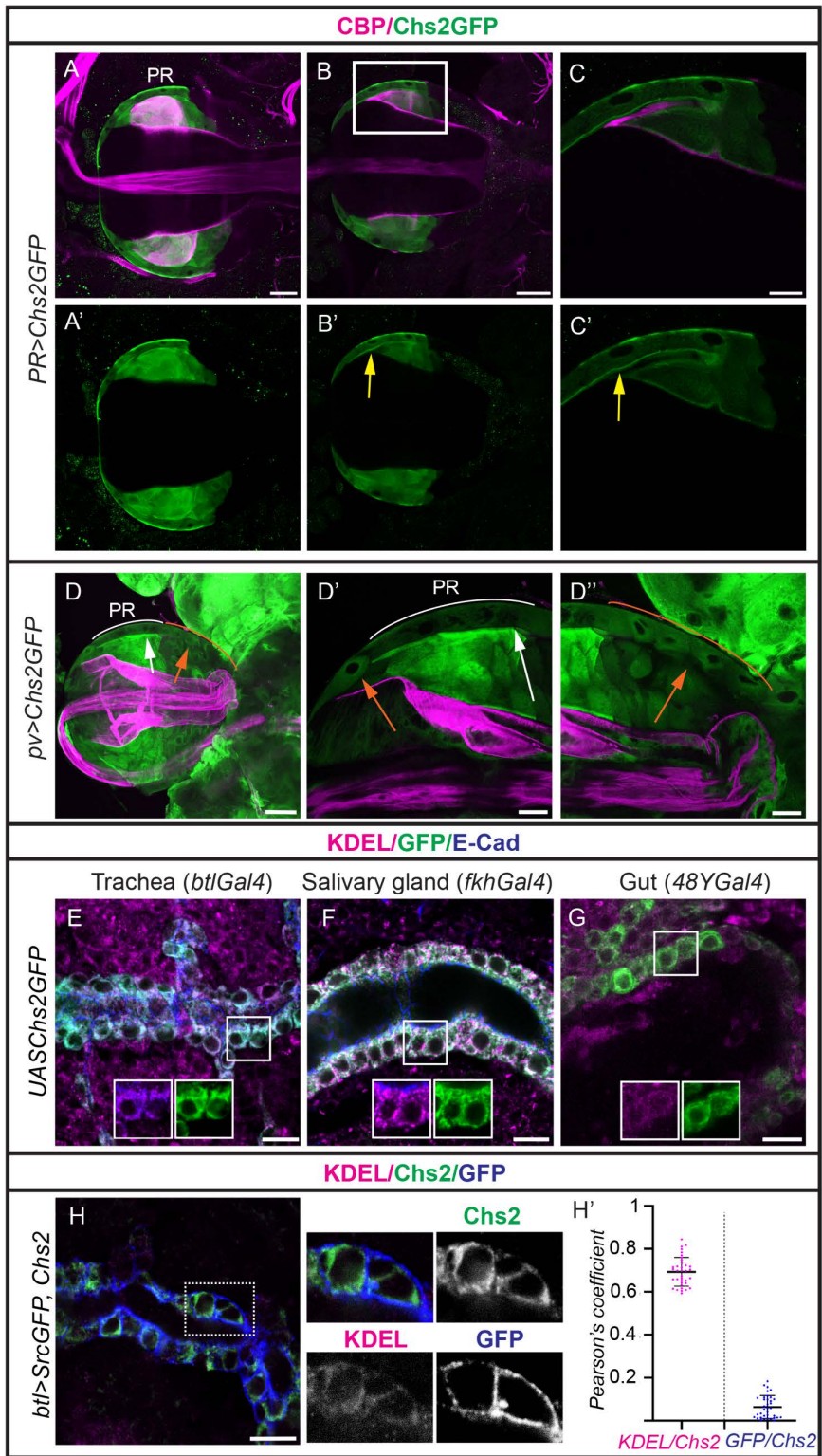

**Fig 5. Subcellular localisation of Chs2.** (A-D) Proventriculus stained for chitin using CBP (magenta) and GFP (green) in larvae expressing Chs2GFP in the PR cells (A-C) or the whole proventriculus (D). A,D correspond to projections and B,C to single sections of A. C corresponds to a magnification boxed in B. Note the accumulation of Chs2GFP in the apical domain of PR cells (yellow arrows), correlating with chitin deposition. Chs2GFP shows

a clear apical localisation only in PR cells (D', white arch and arrow), in spite of being expressed in the rest of midgut cells of the proventriculus (D'', orange arch and arrows). D' and D''corresponds to magnifications of D. (E-G) Single sections of embryos stained for KDEL, GFP to visualise Chs2, and E-Cad to visualise the cells (blue) in the indicated genotypes. Chs2 localises intracellularly, largely colocalising with ER marker in all embryonic tissues analysed (E-G). (H) Single section of embryo stained for KDEL (magenta), Chs2 (green), and GFP to visualise the membrane (blue) in the indicated genotype. Chs2 largely colocalises with KDEL and is not found at the membrane (see magnifications). (H') Scatter plots of the Pearson's correlation coefficient for Chs2/srcGFP and Chs2/KDEL in n = 36 cells analysed from 9 embryos. Scale bars A,B,D 50 μm; C.D',D''20 μm, E-H 10 μm.

Altogether, our experiments indicated that Chs2 primarily accumulates in the ER when it is expressed in ectopic tissues. In contrast, it localises apically in PR cells in which it is endogenously expressed and active. These results suggested that a specific factor/s or condition is present in the PR region to promote ER exit of Chs2.

## 7. Subcellular localisation of Kkv in ectopic tissues

We and others have previously shown that Kkv localises at the apical membrane in the tissues in which it is endogenously expressed [18,19]. When *KkvGFP* is over or ectopically expressed in ectodermal tissues it predominantly localises apically. We also detected *kkvGFP*-containing vesicles, likely reflecting exocytic and endocytic trafficking, as described previously [19]. Additionally, we observed a general cellular distribution, including the ER, likely due to high expression levels (Fig 6A, 6B, S6C, and S6D). KkvGFP subcellular distribution correlated with its ability to deposit chitin extracellulalry in the presence of Exp/Reb or intracellularly in their absence [19,20] (S6H Fig). We then analysed *KkvGFP* when missexpressed in endodermal tissues. In the embryonic gut we detected a diffuse and general pattern of KkvGFP in the cell, with some cortical enrichment but no apical localisation. Double staining with KDEL suggested that a portion of the KkvGFP protein was retained in the ER. However, the presence of KkvGFP-containing vesicles suggested that at least part of the protein can exit the ER and traffic within the cell (Figs 6C and S6F). This correlated with the ability of Kkv to polymerise chitin intracellularly but not to extrude it extracellularly, even in the presence of Reb (Fig 6D, 6E, and S6H). When Kkv was miss-expressed in larval PR cells, in the absence or presence of Reb, a similar diffuse pattern was observed. Again, this correlated with accumulation of chitin intracellularly (Fig 6F, 6G).

Altogether this indicated that Kkv can localise to the apical membrane and deposit chitin extracellularly in ectodermal tissues in the presence of Exp/Reb activity. Conversely, in endodermal derivatives, it does not localise apically, correlating with its inability to deposit chitin extracellularly, even in the presence of Exp/Reb, although it is detected in vesicles and is capable of intracellular chitin deposition.

## 8. Analysis of Chs2-Kkv chimeras

At the sequence level, Chs2 lacks a coiled-coil domain that is present in Kkv. We have proposed that this coiled-coil domain plays a role in protein oligomerisation [19].

To test the relevance of this domain and determine whether this can explain differences between the two CHS, we decided to generate two different GFP-tagged chimeric proteins, one containing Kkv up to the WGTRE domain and Chs2, and a second one containing Chs2 up to WGTRE and Kkv (including the coiled-coil) (kkv-Chs2GFP and Chs2-KkvGFP, respectively, see Fig 7A). We first evaluated their capacity to rescue the absence of chitin in a *kkv* mutant background. We found that none of them was able to restore extracellular chitin deposition or intracellular chitin polymerisation in the trachea of *kkv* mutants (n = 0/11 *kkv* mutant embryos expressing *kkv-Chs2GFP* in tracheal cells deposit chitin, n = 0/18 *kkv* mutant embryos expressing *Chs2-KkvGFP* in tracheal cells deposit chitin). This indicates that none of these chimeric proteins are functional to polymerise and deposit chitin in the ectoderm (Figs 7B–7D). We then tested whether they were able to replace Chs2, and so we expressed them in PR cells in a *Df(Chs2)* background. We found that none of them were able to deposit chitin in the PM (Fig 7E, 7F = 0/7 *Df(Chs2)* escapers expressing *kkv-Chs2GFP* in PR cells deposit PM, n = 0/4 *Df(Chs2)* escapers expressing *Chs2-KkvGFP* in PR cells deposit PM).

**Fig 6. Subcellular localisation of Kkv.** (A-C) Single sections of embryos stained for KDEL (magenta), GFP to visualise Kkv (green) and E-Cad to visualise the cells (blue) in the indicated genotypes. Kkv localises apically in ectodermal tissues (A,B) and diffusely in the endoderm (C). Note, however, the presence of intracellular vesicles indicating trafficking (yellow arrow in C). (D,E) Single sections of the anterior region of the embryo midgut stained for chitin (CBP, magenta), α-spec as a cellular marker (blue) and GFP (green) to visualise Kkv. kkv can deposit chitin intracellularly in the absence or presence of Reb. (F,G) Single sections of proventriculus stained for chitin (magenta) and GFP (green) to visualise Kkv subcellular localisation in PR cells in the indicated genotypes. Kkv accumulates diffusely in the cell in the absence or presence of Reb (yellow arrows). Clear accumulation in intracellular punctae is detected. t, trachea. Scale bars A-C 10 μm, D-G 20 μm.

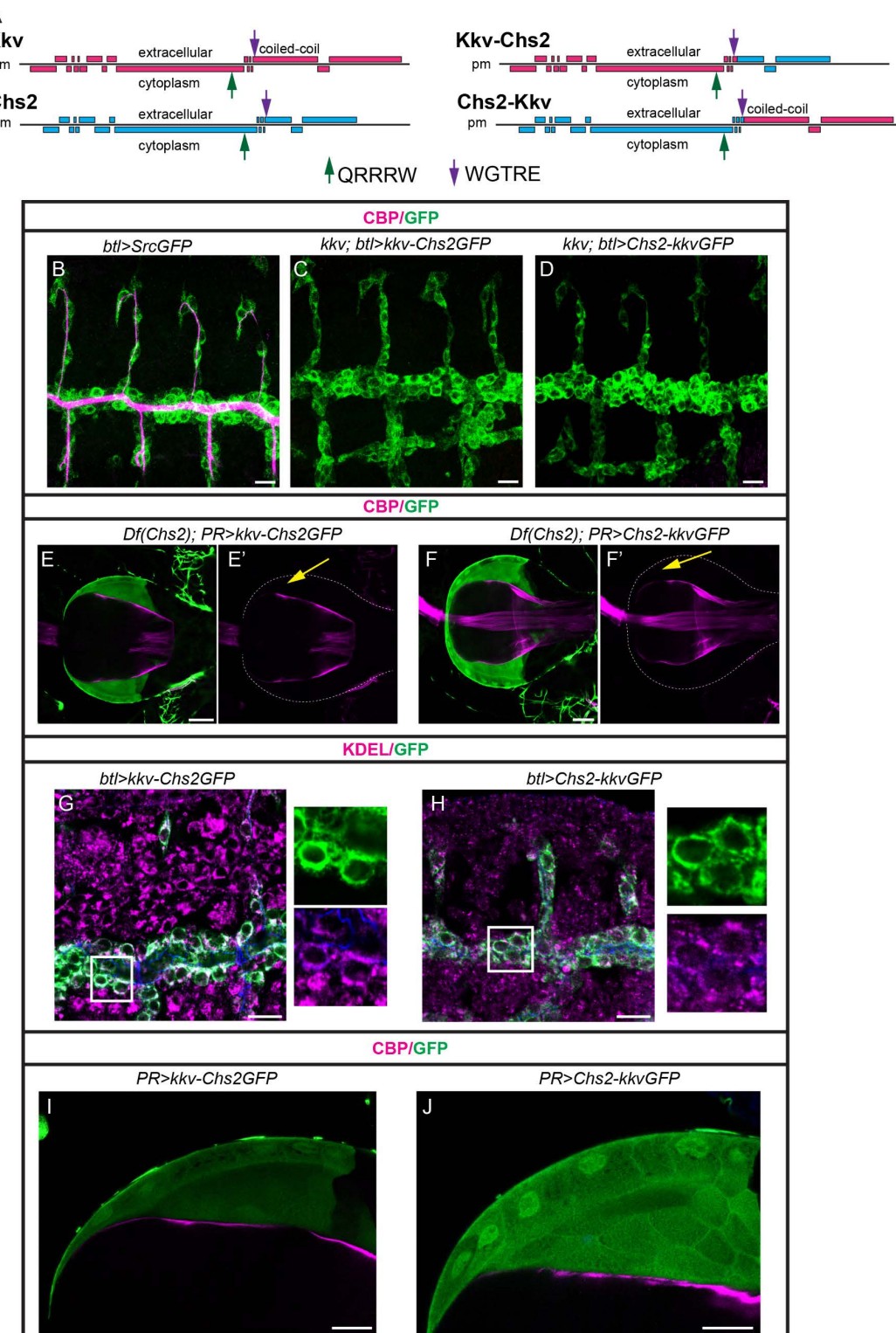

**Fig 7. Kkv and Chs2 chimeras.** (A) Schematics of Kkv and Chs2 showing the different domains and the position of the catalytic domain (QRRRW) and the WGTRE domain. Two different chimeric proteins between Kkv and Chs2 were generated as indicated. (B-J) Embryos and proventriculus of the indicated genotypes are shown, stained as indicated. (B-D) The chimeras cannot rescue the absence of chitin in a *kkv* mutant background. (E-F) The

chimeras cannot rescue the absence of chitin in the PM in a Chs2 mutant background (yellow arrows point to the absence of PM in the proventricular lumen). (G,H) The chimeras localise intracellularly in embryonic tissues, largely colocalising with the ER marker KDEL.(I,J) The chimeras localise diffusely in PR cells. Scale bars B-D,G,H 10μm; E,F,I,J 20 μm.

Finally, we analysed the subcellular accumulation of these chimeric proteins using GFP staining. We found that none of them was able to localise at the membrane in ectodermal tissues, and instead accumulated in the cytoplasm, colocalising with ER markers (Fig 7G, 7H). In the larval proventriculus, expression in the PR cells indicated a diffuse accumulation within the whole cell with no enrichment at the apical domain (Fig 7I, 7J). We also used the Chs2 antibody to visualise the localisation of the chimeras, and found that only kkv-Chs2GFP is recognised by the antibody (S7A and S7B Fig). This result was expected because the antibody was raised against a specific region present in the carboxi-terminal region of Chs2 (see Materials and Methods), which is absent in Chs2-kkvGFP, further validating the antibody's specificity.

Altogether our results indicated that the Kkv-Chs2 chimeras are not able to properly localise at the apical membrane and to polymerise and deposit chitin in ectodermal tissues or in PR cells.

## Discussion

In this work we asked about the specificity of chitin synthases in insects, that typically encode two types of CHS, class A and B, with the exception of hemipteran that encode only one class of CHS. While it is well accepted that class A CHS are expressed and required in ectodermal tissues to deposit chitin in cuticles, and class B in the gut to deposit PM-associated chitin, no studies addressed whether they are functionally equivalent. In fact, most functional analyses are based on downregulation of CHS in different insects, addressing only the requirement of their specific tissular expression (reviewed in [3,4]). Here we leveraged the advantages of functional manipulation offered by *D. melanogaster* to ask whether the two different chitin synthases, Kkv and Chs2, are functionally equivalent. We find that a regulated pattern of expression is not the only factor that underlies CHS specificity. In contrast, our analysis indicates that these two chitin synthases are not functionally equivalent and that they use specific cellular and molecular mechanisms to deposit chitin. We suggest that this specificity found for *D. melanogaster* CHS may also apply for the CHS class A and B in other insects.

### Subcellular localisation of chitin synthases

Chitin synthases contain several transmembrane domains and insert in the plasma membrane. They possess a central domain containing the catalytic domain facing the cytoplasm, and a chitin translocating channel with a gate lock through which the nascent chitin filament is extruded (reviewed in [3]).

The trafficking of CHS to the plasma membrane has been extensively studied, particularly in yeast. In the well-studied case of *Saccharomyces cerevisiae* Chs3, it is known that the protein is synthesised in the ER, where it folds and oligomerises. Chs3 oligomers then exit the ER assisted by the activity of a dedicated chaperone, Chs7. Chs3 is then transported to the Golgi and is finally delivered to the plasma membrane using the exomer complex. Localisation to the membrane also requires the specific activities of different factors, like Chs4 (reviewed in [47]). Interestingly, accumulating data indicate that different CHS use different trafficking strategies. For instance, yeast Chs2 is also synthesised and folded in the ER, but it is retained there in a cell-cycle regulated manner: phosphorylated Chs2 is retained in the ER during mitosis due to high mitotic kinase activity, and is exported to the Golgi after mitosis thanks to the dephosphorylation mediated by Cdc14 [48,49]. Therefore, different post-translational modifications can play key roles in CHS intracellular transport, besides the need of dedicated proteins and chaperones, also in the case of yeast ScChs3 [47]. In any case, the available data indicate that the regulation of intracellular trafficking and remarkably ER exit represents a critical step for yeast CHS activity, which is conserved in fungi [50]. Indeed, reflecting the importance of this specific regulation of intracellular trafficking, CHSs in fungi have been proposed to traffic via chitosomes, which are dedicated intracellular vesicles, distinct from those of the normal secretory route, that serve as a CHS reservoir and trafficking vehicle [51].

Less is known about CHS trafficking in insects, where the existence of chitosomes has not been documented. However, recent advances have highlighted key roles for intracellular trafficking and ER exit in Kkv functional activity [19,23,25]. No prior data existed on *D. melanogaster* Chs2 trafficking. Our studies on Kkv and Chs2 suggest that these distinct CHS undergo specific transport regulation, with ER exit serving as a key regulatory step. We find that Kkv subcellular localisation varies by cell type: in ectodermal tissues, endogenous or overexpressed Kkv primarily localises to the apical membrane, whereas in the endoderm, ectopically expressed Kkv is diffusely distributed within the cell. This pattern suggests that Kkv either requires a dedicated chaperone exclusive to ectodermal tissues or relies on intrinsic ectodermal cell polarity cues for apical membrane localisation. However, in both cell types, Kkv is able to exit the ER, as indicated by the presence of Kkv cytoplasmic vesicles associated with exocytic and endocytic trafficking [19]. In this regard, we showed previously that the WGTRE domain in Kkv is crucial for ER exit; in its absence, we found Kkv in the ER, with no detectable Kkv cytoplasmic vesicles or apical enrichment. Our results suggested that this domain interacts with a chaperone or undergoes an essential posttranslational modification to facilitate ER exit [19]. Chs2 also contains a WGTRE domain, however, our findings strongly suggest that, unlike Kkv, an intact WGTRE domain in Chs2 is not sufficient for ER exit in tissues where it is not endogenously expressed. In these tissues, while we cannot completely rule out that a portion of missexpressed Chs2 exits the ER, it primarily colocalises with ER markers, with no detectable Chs2 cytoplasmic vesicles or apical localisation. This is in sharp contrast with the proventriculus PR cells, where Chs2 is endogenously expressed, and where overexpressed Chs2 primarily localises apically (strongly suggesting that ER retention is not merely a consequence of massive overexpression). Our results suggest the presence of a dedicated chaperone or a specific type of posttranslational modification for Chs2 ER exit and membrane localisation, which is exclusive for PR cells and not sufficient for Kkv. In *Manduca sexta,* Chs2 localises at the apical brush in the midgut of feeding larvae. In this insect, chitin synthase activity in the gut depends on the regulated expression of Chs2. This regulated Chs2 expression, its apical localisation, and posttranslational modifications that activate the synthase activity, are linked to the nutritional state of the larvae [52–54]. Food ingestion, which starts at larval stages, may also contribute to the regulation of Chs2 activity and PM deposition in *D. melanogaster*. However, the observation that Chs2 localises apically in PR cells and not in the rest of proventriculus cells strongly points to the presence of a dedicated ER exit factor or posttranslational modification for *Drosophila* Chs2 ER exit.

In summary, our findings, along those of others, indicate that CHS ER exit regulation is also a critical step in *Drosophila*, pointing to a conserved mechanism across different organisms, including insects and fungi. As in yeast, our results suggest that different CHS in *Drosophila* rely on distinct cellular mechanisms to ensure this critical ER exit step.

Kkv and Chs2 differ in the presence of a coiled-coil domain, which is absent in Chs2. We have recently proposed that this domain plays a role in oligomerisation [19]. In an attempt to further investigate the role of this coiled-coil domain and determine whether it contributes to the specificity of these two chitin synthases, we designed chimeric proteins in which we added the coiled-coil domain to Chs2, or we replaced it in Kkv. We found that these two protein chimeras were primarily localised in the ER. We had previously found that the absence of the coiled coil domain in Kkv does not prevent the protein to exit the ER, reach the apical membrane and to be active [19]. Thus, we speculate that the 2 chimeras we generated are unable to properly fold or oligomerise in the ER, preventing them from reaching the membrane. This suggests that oligomerisation may involve both the N-terminal and C-terminal domains. Oligomerisation is known to be a critical step in yeast CHS. Actually, multiple quality control mechanisms monitor yeast Chs3 folding in the ER. While non-oligomerised Chs3 can exit the ER, they are returned there once they reach the Golgi using a COPI retrograde trafficking, preventing the trafficking of defective molecules [55,56].

## Activity of chitin synthases in *D. melanogaster*

We asked whether Chs2 an Kkv can replace each other. To address this, we first needed to demonstrate that Chs2 deposits chitin in the PM. This analysis allowed us to establish a reliable and easy method to visualise chitin in the PM by

using the widely used chitin probe, CBP. CBP staining revealed chitin deposited at the proventricular lumen and forming a continuous layer along the midgut epithelium, a pattern that provides a valuable experimental assay for further studies of the PM. We also confirmed that Chs2 is the CHS responsible for producing PM-associated chitin, as no chitin was detected in the proventricular lumen or the midgut in its absence. This is an important piece of information that was previously missing in the field.

We found that expressing Chs2 in the PR cells in the proventriculus rescued the absence of chitin in this region under *Df(Chs2)* conditions. Remarkably, our results also demonstrated that the expression of Chs2 exclusively in the PR cells is sufficient to produce the PM chitin deposited in the different intestinal regions, as it also rescued chitin deposition in the layer along the midgut. Moreover, we found that expressing Chs2 in the midgut, but not in the proventriculus, did not restore the chitin layer along the midgut (S2G and S2H Fig). Altogether these findings indicate that, at least at larval stages, the PM lining both the proventriculus and midgut is produced exclusively by PR cells. These results confirm old reports documenting that *D. melanogaster* larvae produce a Type II PM, in which the PM is secreted from a specialised region in the anterior midgut region [34,57]. Thus, the PM is assembled and organised at the proventriculus, presumably structured by a lattice of chitin fibrils, produced by Chs2-expressing PR cells, held by various glycosylated chitin-binding proteins, synthesised by different proventricular cell types. This leads to the formation of a structured PM composed of four distinct layers, each enriched in different proteins (including peritrophins, mucins, chitinases or digestive enzymes) [12,13,32–34]. The PM synthesised in the proventriculus is then poured into the proventricular lumen and pushed posteriorly by a press-like mechanism involving muscular contraction of the proventriculus walls. This ensures that the PM spreads throughout the entire midgut region [34].

We found that Chs2 cannot replace Kkv and rescue the lack of chitin deposition in the trachea in *kkv* mutants, as Kkv does. In addition, it is unable to substitute for Kkv in depositing ectopic chitin in different ectodermal tissues when co-expressed with Exp/Reb. Correlating with these observations, we find Chs2 primarily retained in the ER when expressed in the trachea or other ectodermal tissues. On the other hand, Kkv cannot replace Chs2 when expressed in PR cells, not even in the presence of Reb. Correlating with this observation, we observe a generalised accumulation pattern of Kkv with no apical enrichment, suggesting that it does not localise properly to the apical membrane in the brush border region of PR midgut cells [34]. Altogether, our findings indicate a correlation between CHS subcellular localisation and their ability to deposit chitin. Indeed, insertion of the CHS to the apical membrane is proposed to be a necessary requirement for the translocation of the polymers to the extracellular space, and thus for CHS activity [3,19,58]. Our results suggest the existence of specific cellular trafficking mechanisms for each CHS, ensuring their apical localisation in specific tissues (the ectoderm in the case of Kkv and PR cells in the case of Chs2). In this context, we propose that intracellular trafficking (with ER exit as a key initial regulatory step), and subsequent subcellular localisation (with insertion into the apical membrane as a second critical step), play essential roles in regulating CHS activity [19]; this work). We do not rule out the possibility that CHS require distinct enzymatic activities and/or cofactors for chitin polymerisation/deposition, which may vary depending on the cell type. Indeed, auxiliary proteins such as Exp/Reb are specifically expressed in certain ectodermal tissues [20]. These mechanisms could act jointly or in parallel with the regulation of intracellular trafficking, or could even regulate this intracellular trafficking itself. Identifying the exact mechanisms controlling Kkv and Chs2 intracellular trafficking would be necessary to determine whether additional mechanisms (specific cofactors or enzymatic activities) are also involved, or possibly act as the primary regulatory elements.

Our results also suggest different molecular mechanisms for chitin deposition by Kkv and Chs2. While Kkv requires the function of Exp/Reb [19,20], Chs2 appears to function independently of this activity. Notably, downregulation of Exp/Reb in PR cells, where Chs2 is normally active, does not cause detectable defects in PM-chitin deposition. Furthermore, we observed no differences in the effects caused by Chs2 expression across the different tissues analysed (trachea, salivary glands, midgut or proventriculus) when Reb is also present. scRNAseq revealed no expression of *reb/exp* in the midgut cells of the adult proventriculus [32], and we similarly detected no expression of *reb/exp* in PR cells at larval stages. However, we did observe *exp* expression in the ectodermal cells of the larval proventriculus (S4C Fig), which correlates

with *kkv* expression in that region and with cuticle deposition. Based on these findings, we propose that in contrast to ectodermal Kkv, Chs2 likely relies on a distinct set of auxiliary proteins and uses a different mechanism to regulate the chitin-translocating channel. We note that further work will be needed to determine whether other previously identified Kkv auxiliary proteins, such as SERCA or Knk, are required for Chs2-mediated chitin deposition, and to identify putative specific auxiliary proteins for Chs2 chitin depostion. Whether the mechanistic differences result in distinct structural organisation of chitin promoting specific functions in the cuticles versus the PM remains to be investigated. In this context, the ability to visualise PM-associated chitin using CBP staining is informative, as CBP recognises insoluble or crystalline chitin but not chito-oligosaccharides or insoluble derivatives of chitin [59]. While the structural organisation of chitin in cuticles is well studied, it should be noted that there is a lack of structural analyses focused on understanding chitin organisation and its modifications in the PM of different insect species [12,14].

In summary, our work demonstrates that that the activity and specificity of *D. melanogaster* CHS depend on multiple layers of regulation. These include the spatio-temporal regulation of their expression pattern, regulation of intracellular trafficking, ER exit and membrane insertion, as well as the the organisation of specific molecular machineries for chitin deposition. Additionally, our work lays the grounds and provides experimental frameworks for future studies aimed at elucidating the physiological role of the PM, identifying the specific molecular complexes involved in chitin deposition by different CHS, and exploring the structural organisation and properties of chitin fibers deposited by different CHS. These aspects are key for a comprehensive understanding of the diverse functional requirements and potential biotechnological applications of chitin and chitin-modified polymers.

## Materials and methods

### *D. melanogaster* strains and maintenance

All *D. melanogaster* strains were raised at 25ºC under standard conditions. Balancer chromosomes were used to follow the mutations and constructs of interest in the different chromosomes. For overexpression experiments, we used the Gal4 drivers *btlGal4* (in all tracheal cells, kindly provided by S. Hayashi *RRID* FBal0052380), *PRGal4* (in PR cells, CG15,153$^{CR02499-TG4.2}$, *RRID* BDSC#92,290), *pvGal4* (in all proventriculus, Gnpnat$^{CR70964-TG4.0}$, *RRID* BDSC#600,028), *fkh-Gal4* (in salivary glands, kindly provided by D. Andrew, *RRID* BDSC#78,060), *48YGal4* (in endoderm, *RRID* BDSC#4,935) and *Mex1Gal4* (in enterocytes, *RRID* BDSC#91,368). The overexpression and rescue experiments were performed using the Gal4/UAS system [60].

The following fly strains were used: *kkv$^{lB22}$* (*RRID* FBal0179290), *UASChs2 and UASKkvGFP* (*RRID* FBal0302989) (kindly provided by B. Moussian). The following stocks were obtained from Bloomington *D. melanogaster* Stock Center (BDSC): y$^1$w$^{1,118}$ (*RRID* BDSC#6,598, used as control), *Df(3L)BSC233 and Df(3L)BSC284* (*RRID* BDSC#9,700 and BDSC#23,669 respectively, deficiencies for *Chs2*), *UAS-reb* (*reb$^{LA0073}$, RRID* BDSC#22,192), *UAS-rebRNAi* (*RRID* BDSC#35,031), Chs2$^{CR60212-TG4.0}$ (*RRID* BDSC#93,534), kkv$^{smGFP-HA}$ (*RRID* BDSC#84,993), exp$^{CR60016-TG4.2}$ (*RRID* BDSC#93,301), fkh$^{GFP}$(*RRID* BDSC#43,951, to visualise Fkh pattern), *UASnlsCherry* (*RRID* #38,425) and *UASsrcGFP* (RRID BDSC#5,432, used as membrane marker; the myristylation domain of v-src fused to EGFP targets it to the plasma membrane) and Vienna *Drosophila* Resource Center: *UAS-expRNAi* (*RRID* VDRC#102,370). The UASChs2GFP, UAS-GFP-kkv-Chs2 and UAS-GFP-Chs2-Kkv were generated in our lab.

### Immunohistochemistry

Third instar larvae were dissected in cold PBS and then fixed in 4% Paraformaldehyde for at least 2 hours at room temperature (RT). Larvae were washed 3x15min in PBS-0.3% Triton X-100 (PBT). Dissected guts were blocked with PBT-BSA for 30 min and then incubated with the primary antibodies in PBT-BSA overnight at 4ºC. Guts were washed with PBS 3x15min and incubated with secondary antibodies in PBT-BSA at RT for 2–4 hours in the dark, washed in PBS and mounted in Vectashield (Vector Laboratories, H1000-10) on microscope glass slides and covered with thin glass slides.

Embryos were stained following standard protocols. Embryos were fixed in 4% formaldehyde (Sigma-Aldrich, 103,999) in PBS1x-Heptane (1:1) for 10 min for E-cad staining and for 20 min for the rest. Embryos transferred to new tubes were washed in PBT-BSA blocking solution and shaken in a rotator device at room temperature. Embryos were incubated with the primary antibodies in PBT-BSA overnight at 4ºC. Secondary antibodies diluted in PBT-BSA (and for the CBP staining) were added after washing and were incubated at room temperature for 2–5 h in the dark. Embryos were washed, mounted on microscope glass slides and covered with thin glass slides.

The following primary antibodies were used: goat anti-GFP (1:600, AbCam, AB_3373); rabbit anti-GFP (1:600, ThermoFisher Scientific – Invitrogen, A11122); mouse anti-KDEL (1:200, Stressmarq Biosciences, SMC-129D), mouse anti-FK2 (1:50, Enzo Life Science, ENZ-ABS840–0100), rat anti-E-Cadh, DCAD2 (1:100, Developmental Studies Hybridoma bank-DSHB, DSHB#528,120), mouse anti-α-Spec (1:10 DSHB#528,473), mouse anti-mCherry-3A11 (1:100 DSHB#2,617,430), rabbit anti-Dve (1:5, kindly provided by Dr. H. Nakagoshi) and rat anti-Chs2 (1:100, this work).

Cy3-, Cy2- and Cy5-conjugated secondary antibodies (Jackson ImmunoResearch) were used at 1:300. Chitin binding probe fluorescently labelled CBP (1:300) was used to visualise chitin (prepared by N. Martín)

### Image acquisition

Fluorescence confocal images of fixed embryos and digestive tracts were obtained with Leica TCS-SPE system using 20x and 63x (1,40−0,60 oil) objectives (Leica). Fiji (ImageJ) [61] was used for adjustment. Confocal images are maximum-intensity projections of Z-stack sections or single sections as indicated in the the legends. Figures were assembled with Adobe Illustrator.

### Colocalisation analysis

Images were acquired as described above. Before the colocalisation analysis, images were pre-processed using the Fiji software. The images were splitted into three channels corresponding to Chs2, the membrane markers srcGFP and the ER marker KDEL. For each channel, a median filter with a radius of 10 was applied to reduce noise. To enhance the contrast and remove the background, the original image was substracted from the filtered image using the Image Calculator function. The processed images were analysed. With the Fiji software, a ROI was drawn to select a single cell from a single stack. The selected cell was duplicated for all processed channels. Colocalisation analysis was performed using the JACoP tool, calculating the Pearson's correlation coefficient (r) to evaluate the colocalisation between Chs2 and srcGFP as well as Chs2 and KDEL. r ranges between −1 and 1, where 1 indicates perfect correlation, 0 no correlation, and −1 perfect anti-correlation. Typically, an r value of 0.7 and above is considered a strong positive correlation, whereas a value below 0,1 is regarded as very weak or no correlation.

### Generation of UAS constructs

Generation of UAS-GFP-Chs2: We amplified the GFP sequence through PCR (Q5 Hot Start High-Fidelity 2X Master Mix, NEB #M0494S), in the sense primer we included at 5′ the restriction site for XbaI and the Kozak sequence and in the anti-sense primer at 3′ the restriction sites for NdeI-NheI-XbaI. Upon digestion of the standard cloning vector SK with XbaI, we cloned the amplified sequence of GFP in the plasmid. Chs2 was amplified through PCR and we added at 5′ the restriction site for NdeI and at 3′ the restriction site for NheI. The amplified DNA was cloned into SK-GFP upon digestion of the construct with NdeI/NheI. UAS-GFP-Chs2 (Chs2GFP) was obtained digesting SK-GFP-Chs2 with the restriction enzyme XbaI and cloned in pUAST-attB vector. Upon ligation of the fragments, *E. coli* competent cells were transformed and plated in selective plates. Miniprep to obtain DNA were performed using the kit NZYtech (#MB01002) and the DNA was sequenced through the platform Eurofins Genomics. After performing a midiprep (NZYtech, #MB05004/5), the DNAs were injected in embryos y[1]w[1,118] by the "Drosophila injection Service" of the "Institute for Research in Biomedicine" (IRB, Barcelona) and by the "Transgenesis Service" of the "Centro de Biología Molecular Severo Ochoa" (CBM, Madrid).
The primers used are the following.

PCR primers to amplify GFP: sense 5′- GCT CTA GAG ATG GTG AGC AAG-3′ and antisense 5′-CGT CTA GAA GGC CTG CTA GCC ATA TGC TTG TAC AGC TCC TC-3′;

PCR primers to amplify ChS2: sense 5′-GGA ATT CCA TAT GAG CGG AGG AGAC-3′ and antisense 5′-CTA GCT AGC GCT CTC TGG CGC GTG GGT-3′.

Generation of UAS-GFP-kkv-Chs2 and UAS-GFP-Chs2-Kkv: The constructs UAS-GFP-kkv-Chs2 and UAS-GFP-Chs2-Kkv were obtained by recombination of specific fragments of DNA using the kit "NEBuilder HiFi DNA Assembly" (NEB #E5520S). The recombination occurs between two sequences of identical DNA, for this reason when we amplified the regions of interest through PCR (Q5 Hot Start High-Fidelity 2X Master Mix, NEB #M0494S), we added a nucleotide tail complementary to the sequence of the vector and a second tail complementary to the sequence of the other fragment. The DNA templates used during the PCR were pJET1.2-GFP-kkv [19] and SK-GFP-ChS2 (described above). We also used the pUAST-attB vector linearised by the restriction enzymes XhoI/KpnI. The kit comprehended material to obtain seamless assembly of multiple DNA fragments. Upon recombination of the fragments, we proceeded as described above with the transformation of *E. coli* competent cells, DNA extraction through miniprep, sequencing of the DNA, DNA extraction through midiprep and, finally, injection of *D. melanogaster* embryos.

The primers used in this study are the following:

PCR primers to amplify GFP-kkv from 5′ until WGTRE motif, with tails overlapping pUAST-attB and Chs2 after WGTRE motif: sense 5′- CTG CGG CCG CGG CTC GAG GGT ACC TCT AGA TGG TGA GCA GGG CGG AGG AG-3′ and antisense 5′-TGA GCA CTG GAG CCT CGC GGG TGC CCC AGGA-3′;

PCR primers to amplify Chs2 after WGTRE motif until 3′, with tails overlapping pUAST-attB and GFP-kkv until WGTRE motif: sense 5′-GGG CAC CCG CGA GGC TCC AGT GCT CAA GGAC-3′ and antisense 5′-AGT AAG GTT CCT TCA CAA AGA TCC TCT AGA ACT CGG TGT GCTC-3′;

PCR primers to amplify GFP-Chs2 from 5′ until WGTRE motif, with tails overlapping pUAST-attB and kkv after WGTRE motif: sense 5′-TCG TTA ACA GAT CTG CGG CCG CGG CTC GAG ATG GTG AGC AAG GGC GAG-3′ and antisense 5′-TCT TAG CCA CCA CCT CTC GAG TGC CCC ACG AAA AC-3′;

PCR primers to amplify kkv after WGTRE motif until 3′, with tails overlapping pUAST-attB and GFP-Chs2 until WGTRE motif: sense 5′-GGG CAC TCG AGA GGT GGT GGC TAA GAA GAC CAA GAA AG-3′ and antisense 5′-CTT CAC AAA GAT CCT CTA GAG GTA CCT CAC AGG CGA CCT GTG CC-3′.

## Generation of antibodies

To generate a polyclonal antibody against Chs2, fragments were amplified by PCR using the following primer combination: sense 5′- CATGCCATGGACTGGTTTCGAACAGGAGGT −3′ and antisense 5′- CCGGAATTCAAAGCCATATTGTTCCG −3′. This amplified a fragment aa F1222 to aa L1383, with low homology to Kkv. The amplified fragments were cloned into the expression vector pPROEXHTa (NcoI/EcoRI). The construct was transformed into Rosetta(DE3) competent cells, and a selected positive clone was induced with 1 mM IPTG at 37ºC during 2 hours. The expressed 20 KDa protein fused with a His tag was purified through a column of Ni-NTA (Thermofisher #88,221) in denaturalising conditions (8 M urea). The purified protein was used to inject rabbits by the facility CID-CSIC-Production of antibodies (Barcelona).

## Supporting information

**S1 Fig. Related to Fig 1.** (A) Schematics of a sagittal section of half proventriculus indicating the position of the APR,PR and PPR midgut cells. (B) Confocal projection of a wild type proventriculus of a L3 larva stained for chitin (CBP, magenta) and α-Spec as cellular marker (green). PR, APR and PPR cells are indicated. Note the enrichment of chitin in the region of PR cells, corresponding to the PM. (C-E) Chitin deposition in relation to ectodermal/endodermal regions. The Fkh pattern (green) marks the ectodermal cells, which deposit a cuticle along their apical domain (magenta) (C,D). Dve marks

the endodermal cells (blue), which deposit a PM along their apical domain (yellow arrow in C,D). In conditions of Chs2 absence, no PM is detected in the endodermal regions (yellow arrow in E), but ectodermal cells still deposit a cuticle (E). Scheme in D indicates the apical and basal domains in the different regions in the proventriculus. (C) Schematics representing the chromosomal region of Chs2 and the deficiencies used to remove the gene. The combination of the deficiencies gives rise to L3 scapers that lack chitin enrichment in the proventriculus and lining the midgut (yellow arrows and magnification of boxed region in grey), compared to their sibling heterozygote larvae (white arrows point to PM, magnification of boxed region in grey). Scale bars C,E,F-proventriculus 50 µm; F-midgut 20 µm.
(TIF)

**S2 Fig.  Related to Fig 2.** (A,B) Montage of several confocal projections stained for GFP (green) to include the whole digestive tract. Note that the *PRGal4* line used is expressed only in the PR cells and in the hindgut. Mex1 is expressed in the midgut and is absent in the proventriculus. (C-H) L3 digestive tracts stained for chitin using CBP (magenta) and α-Spec in the indicated genotypes. All images correspond to single confocal sections. No chitin in the PM in L3 *Df(Chs2)* escapers is observed (yellow arrows in C,D). Expression of *Chs2* in PR cells rescues chitin deposition in the PM (white arrows in E,F). Expression of *Chs2* in the midgut does not rescue chitin deposition in the PR region or along the midgut (yellow arrows in G,H). Scale bars A 200 µm; C,E,G 50 µm; C',E',G',D,F,H 20 µm.
(TIF)

**S3 Fig.  Related to Fig 3.** (A-D) Confocal projections showing dorso-lateral views of the trachea stained for chitin (CBP, magenta), GFP to visualise the tracheal cells (blue) and E-Cad (green). A luminal filament assembles at early stages (A) and a cuticle with the taenidial pattern forms later (B) in the wild type. Chitin deposition is not rescued in *kkv* mutants when adding back *Chs2* (C,D). (E-H) Single sections (E-G) or projections (E'-G', H,H') of salivary glands stained for chitin or Kkv (magenta) and to highlight the salivary glands (green) in the indicated genotypes. Chs2 cannot deposit chitin intracellularly when expressed alone (E). In combination with Reb (F) there is some deposition of chitin in the lumen (as when expressing Reb alone, Fig 3N). In *kkv* mutants, the expression of Reb cannot promote the deposition of chitin in salivary glands (G). This indicates that the luminal staining in *fkh>reb* conditions is due to the presence of Kkv. Accordingly, Kkv protein is detected in the salivary glands (H). (I) Single sections of proventriculus and midgut stained for chitin (CBP, magenta) and α-Spec as cellular markers (green). The concomitant downregulation of *exp* and *reb* in PR cells does not prevent chitin deposition in the proventriculus (yellow arrow in I) and midgut (yellow arrow in J, magnification of boxed region in grey). Chitin is also detected in intracellular punctae in PR cells (blue arrow in I). Scale bars A-H 10 µm; I 50µm;J 20 µm.
(TIF)

**S4 Fig.  Related to Fig 4.** Expression pattern of *Chs2* in the embryo and proventriculus and *exp* in the proventriculus. (A) Confocal projection of Fkh$^{GFP}$; *Chs2$^{CR60212-TG4.0}$-UASnlsCherry* proventriculus stained for GFP (green, to visualise Fkh), Cherry (magenta, to visualise Chs2 pattern) and Dve (blue). Chs2 colocalises with Dve and is absent from the ectodermal region. (A,B) Single sections of embryos at stage 16 stained with α-Spec (magenta) to visualise the cells and GFP to visualise Chs2 pattern. Note the expression in a few cells in the proventriculus. B corresponds to a magnification of A. (C) Single sections of proventriculus stained for chitin (magenta), GFP (green) to visualise Exp expression and α-Spec (blue) to visualise the cells. Exp is expressed in the ectodermal region of the proventriculus and the esophagus and it is absent in PR cells. ecto-ectoderm; eso-esophagus Scale bars A,B,D 50 µm; C,D' 20 µm
(TIF)

**S5 Fig.  Related to Fig 5.** (A,B) Single sections of proventriculus stained for chitin (magenta) and GFP (green) to visualise Chs2 subcellular localisation in PR cells in the indicated genotypes. Chs2 is enriched at the apical domain of PR cells (yellow arrows in A', B') in the absence or presence of Reb, and its accumulation correlates with a clear enrichment of chitin. Note the fibrous aspect of chitin in the PM (blue arrows in A'', B''). (C-E) Single sections of embryos stained for KDEL

(magenta), anti-Chs2 to visualise Chs2 in the indicated genotypes. Chs2 localises intracellularly, largely colocalising with the ER marker in all embryonic tissues analysed. (F) Single section stained for Fk2 (magenta), GFP to visualise Chs2 (green), and E-Cad to visualise the cells (blue). Chs2GFP does not colocalise with FK2. Scale bars A, B 20 μm, C-F 10 μm.
(TIF)

**S6 Fig.  Related to Figs 5 and 6.** Confocal projections of embryonic trachea or intestinal tracts (at the indicated stages) upon *Chs2GFP* or *KkvGFP* overexpression using *btlgal4* (expressed in tracheal cells) or *48YGal4* (expressed in the ectodermal-foregut and endodermal-midgut). Embryos are stained for the indicated antibodies. In the trachea, KkvGFP can be found in the whole cell, particularly at early stages (C), but it is predominantly localised apically (blue arrows in C,D) and detected in intracellular vesicles (yellow arrows C,D). In contrast, Chs2GFP localises in the whole cell (A,B). In the intestinal tract, KkvGFP localises apically in the foregut region (f, encircled in white, blue arrow in F') and it is diffusely localised within the cell in the midgut (m). In both regions, KkvGFP is detected in vesicles (yellow arrows in F). The pattern correlates with the ability of Kkv to deposit chitin extracellulaly in the foregut (encircled in white, orange arrow in H') or intracellularly in the midgut (red arrow in H'). In contrast, Chs2 localises diffusely in the whole cell in the foregut (encircled in white in E, G) or midgut regions. tr, trachea; f, foregut; m, midgut. Scale bars 10 μm.
(TIF)

**S7 Fig.  Related to Fig 7.** (A,B) Confocal projections showing dorso-lateral views of the trachea stained for GFP (green), Chs2 antibody generated in the lab (blue, see Materials and methods) and E-Cad (magenta) to follow the tubes in the indicated genotypes. Both chimeras are tagged with GFP (A', B'). Only *kkv-Chs2GFP* is recognised by Chs2 antibody (yellow arrow in B''), which was generated against a region in the Carboxi-terminal domain of Chs2 that is not present in *Chs2-kkvGFP* (yellow arrow in A''). Scale bars 20 mm.
(TIF)

**S1 File.  Phenotypic analysis.** Table indicating the genotype of the fly lines crossed to obtain the genetic combinations analysed. The phenotypic character analysed is indicated (e.g. deposition of chitin in the PM, trachea, etc). The penetrance of the phenotype is indicated as the number of embryos/larvae of the indicated genetic combination showing the phenotype with respect the total number of embryos/larvae analysed.
(XLSX)

**S2 File.  Colocalisation analysis.** Pearson's correlation coefficient data evaluating the colocalisation between Chs2-srcGFP and Chs2-KDEL for each cell analysed (n = 36 cells from 9 independent embryos). Statistical analysis (*t* test with Welch's correction) indicates clear distinct pattern of accumulation of Chs2, mainly in the ER. These data refer to Fig 5H.
(XLSX)

## Acknowledgments

We thank L. Portas and C. Rojano for contributions to the initial stages of this project and A. Letizia and ML. Espinàs for continuous help. We also thank J. Geble and A. Wodarz for help and advice. We thank the IBMB Molecular Imaging Platform for technical help. We acknowledge the Bloomington Stock Centre, Vienna Drosophila Resource Center and the Developmental Studies Hybridoma Bank for fly lines and antibodies. We thank B. Moussian, S. Hayashi, H. Nakagoshi and D. Andrew for kindly providing fly lines and antibodies. We thank the members of the Llimargas and Casanova labs and S. Araújo for helpful discussions, and A. Letizia, ML. Espinàs, J. Casanova and X. Franch-Marro for critical reading of the manuscript. N. Martín is a technician in Prof. J. Casanova's lab.

## Author contributions

**Conceptualization:** Marta Llimargas.

**Formal analysis:** Joan Bertran-Mas, Ettore De Giorgio, Marta Llimargas.

**Funding acquisition:** Marta Llimargas.

**Investigation:** Joan Bertran-Mas, Ettore De Giorgio, Marta Llimargas.

**Methodology:** Joan Bertran-Mas, Ettore De Giorgio.

**Resources:** Nicolás Martín.

**Supervision:** Marta Llimargas.

**Validation:** Joan Bertran-Mas, Ettore De Giorgio, Marta Llimargas.

**Writing – original draft:** Marta Llimargas.

**Writing – review & editing:** Joan Bertran-Mas, Ettore De Giorgio, Marta Llimargas.

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
