## [Decision Letter · Decision Letter 0]

9 Jul 2025

PGENETICS-D-25-00587

Distinct cellular and molecular mechanisms contribute to the specificity of the two Drosophila melanogaster chitin synthases in chitin deposition.

PLOS Genetics

Dear Dr. Llimargas,

Thank you for submitting your manuscript to PLOS Genetics. After careful consideration, we feel that it has merit but does not fully meet PLOS Genetics's publication criteria as it currently stands. Therefore, we invite you to submit a revised version of the manuscript that addresses the points raised during the review process.

Please submit your revised manuscript within 30 days Aug 08 2025 11:59PM. If you will need more time than this to complete your revisions, please reply to this message or contact the journal office at plosgenetics@plos.org. Please include the following items when submitting your revised manuscript:

We look forward to receiving your revised manuscript.

Kind regards,

Ken M. Cadigan, PhD

Academic Editor

PLOS Genetics

Fengwei Yu

Section Editor

PLOS Genetics

Aimée Dudley

Editor-in-Chief

PLOS Genetics

Anne Goriely

Editor-in-Chief

PLOS Genetics

**Additional Editor Comments :**

I would like you to address the minor criticisms of reviewers' 2 & 4, which do not require additional experiments but are intended to benefit the readership. But you can ignore reviewer 2's comments about deleting the chimeras.

**Journal Requirements:**

https://journals.plos.org/plosgenetics/s/submission-guidelines#loc-parts-of-a-submission

3) Some material included in your submission may be copyrighted. According to PLOSu2019s copyright policy, authors who use figures or other material (e.g., graphics, clipart, maps) from another author or copyright holder must demonstrate or obtain permission to publish this material under the Creative Commons Attribution 4.0 International (CC BY 4.0) License used by PLOS journals. Please closely review the details of PLOSu2019s copyright requirements here: PLOS Licenses and Copyright. If you need to request permissions from a copyright holder, you may use PLOS's Copyright Content Permission form.

Potential Copyright Issues:

i) Figure 1B. Please confirm whether you drew the images / clip-art within the figure panels by hand. If you did not draw the images, please provide (a) a link to the source of the images or icons and their license / terms of use; or (b) written permission from the copyright holder to publish the images or icons under our CC BY 4.0 license. Alternatively, you may replace the images with open source alternatives. See these open source resources you may use to replace images / clip-art:

4) We note that your Data Availability Statement is currently as follows: "All relevant data are within the manuscript and its Supporting Information files.". Please confirm at this time whether or not your submission contains all raw data required to replicate the results of your study. Authors must share the “minimal data set” for their submission. PLOS defines the minimal data set to consist of the data required to replicate all study findings reported in the article, as well as related metadata and methods (https://journals.plos.org/plosone/s/data-availability#loc-minimal-data-set-definition).

2) If any authors received a salary from any of your funders, please state which authors and which funders..

**Reviewers' comments:**

Reviewer's Responses to Questions

Reviewer #1: well done, no further comments.

Reviewer #2: Previous reviews of this manuscript were detailed and comprehensive. I fully agree with both reviewers’ assertions that the work is important and beautifully executed. For the most part, the authors addressed the reviewers’ main points nicely in this revised version (but see below). I will be interested to see what their work on physiological mechanisms reveals in future submissions. I do wonder if in their experiment #2 to address Reviewer #1's point 3, where they starved larvae, they were able to detect a change in the deposition of chitin in the midgut (Reviewer #2's point 4). If one of the ideas for how the PM is distributed is from food pushing through the gut, one might expect it to be less organized in the starved larvae.

The authors may have misunderstood Reviewer 2’s points 3 and 5, which asks them to quantify the restoration of chitin in mutants. I suspect that the reviewer was asking if the amount of chitin was fully restored, not how many animals were examined and showed detectable deposition of chitin. In their rebuttal, the authors use terms like “full rescue” and “100% restoration” when they just mean that chitin was detected, not that the amount detected is equivalent to the wild-type level. While documenting the penetrance is helpful, they should not use absolute terms that imply quantification of amount when they are simply measuring presence vs. absence of chitin-binding protein fluorescence. I note that on page 7 the authors state “Chs2 expression was able to restore the chitin accumulation in the PM” which implies full restoration. This could be softened to “was able to restore detectable chitin accumulation”. On page 9, they state “expression of kkv in tracheal cells of kkv mutants fully restores the accumulation of chitin in the luminal filament and in the tracheal cuticle (100% rescue, n=27/27 …”. Again, the term “fully restores” is not appropriate if the absolute amount of chitin is not measured. Finally, in the revised discussion they state “expression of Chs2 exclusively in the PR cells is sufficient to produce all the chitin deposited in the PM”, where saying “all” may not be accurate since the amount was not quantified.

My only remaining question about this manuscript is whether the experiments analyzing the chimeric molecules should be included. Neither chimeric construct is correctly localized, and both appear to be retained in the ER. This suggests that the novel molecules fail to fold properly or have some internal incompatibility that causes aggregation. The negative results presented in Fig. 7 do not seem to contribute anything meaningful to this paper. Perhaps in future a more surgical approach – inserting the coiled-coil domain alone into the Chs2 sequence or swapping the multi-TM-spanning domains in the amino-termini – might be more informative.

Overall, I found the data nicely support the authors’ conclusions and the paper represents a solid advance in our understanding of chitin deposition in insects.

Minor comments:

In fig. 1B, it might be helpful to make a dotted line box outlining the portion of the proventriculus being represented in fig. 1C.

Typos:

Pg. 4 “in a specific a region of the proventriculus”

Pg 7 “penentrant”

Pg. 18 “glycosilated”

Pg. 31 “shematic” (twice)

Reviewer #3: This is an exciting article that address a very important but neglected topic, the role of Chitin synthase. The paper is complete.

Reviewer #4: In their manuscript entitled “Distinct cellular and molecular mechanisms contribute to the specificity of the two Drosophila melanogaster chitin synthases in chitin deposition” the authors Joan Bertran-Mas and colleagues report on their findings on molecular aspects of chitin synthesis occurring in two developmentally separate tissues, the ectoderm and the endoderm by two tissue-specific enzymes. They find that the cellular mechanisms of chitin synthesis adjustment including the control of trafficking through the ER might be similar but that specific partner proteins are employed to implement these routes. The apply strong genetic and histological methods to tackle the problem and elegantly characterize it; however, they do not provide molecular evidence for this observation. Nevertheless, their data, for the first time, shed some light on an extremely complex cell biological issue that surely will interest colleagues outside the insect biology field and provide new possibilities to continue in this direction.

The manuscript was submitted along with two full reviews communicated by Review Commons. These reviews cover the subject in detail and served to improve the presentation of the work considerably. Besides the points raised by the reviewers, I would like to propose the following three points to be addressed in addition:

1) The introduction does not give enough information on the data and concept of chitin synthesis machinery available for insects, where several proteins mentioned in the results and discussion section are known to assist the central enzyme chitin synthase (mainly the epidermal one: SERCA, Reb, Knk/Rtv etc). This information should be given in the introduction and point by point discussed in the discussion.

2) In the same direction, the concept of chitosomes in fungi/yeasts should/could be introduced and discussed as comparison with the fungal system has already been included in the manuscript.

3) Ther type 2 peritrophic matrix (PM) has been proposed to issue from chitin synthase activity in the PR cells only. To follow the argument, a more detailed description of PM types should be given.

**Have all data underlying the figures and results presented in the manuscript been provided?**

Reviewer #1: Yes

Reviewer #2: Yes

Reviewer #3: Yes

Reviewer #4: Yes

PLOS authors have the option to publish the peer review history of their article (what does this mean? ). If published, this will include your full peer review and any attached files.

**Do you want your identity to be public for this peer review?** For information about this choice, including consent withdrawal, please see our Privacy Policy .

Reviewer #1: **Yes: ** Matthias Behr

Reviewer #2: No

Reviewer #3: No

Reviewer #4: No

**Figure resubmission:**
---

## [Editor Report · Decision Letter 1]

21 Aug 2025

Dear Dr Llimargas,

We are pleased to inform you that your manuscript entitled "Distinct cellular and molecular mechanisms contribute to the specificity of the two Drosophila melanogaster chitin synthases in chitin deposition." has been editorially accepted for publication in PLOS Genetics. Congratulations!

Yours sincerely,

Ken M. Cadigan, PhD

Academic Editor

PLOS Genetics

Fengwei Yu

Section Editor

PLOS Genetics

Aimée Dudley

Editor-in-Chief

PLOS Genetics

Anne Goriely

Editor-in-Chief

PLOS Genetics

Comments from the reviewers (if applicable):

**Data Deposition**

http://datadryad.org/submit?journalID=pgenetics&manu=PGENETICS-D-25-00587R1

**Press Queries**

---

## [Editor Report · Acceptance letter]

PGENETICS-D-25-00587R1

Distinct cellular and molecular mechanisms contribute to the specificity of the two Drosophila melanogaster chitin synthases in chitin deposition.

Dear Dr Llimargas,

We are pleased to inform you that your manuscript entitled "Distinct cellular and molecular mechanisms contribute to the specificity of the two Drosophila melanogaster chitin synthases in chitin deposition." has been formally accepted for publication in PLOS Genetics! Your manuscript is now with our production department and you will be notified of the publication date in due course.

With kind regards,

Anita Estes

PLOS Genetics

On behalf of:
